# Exploiting Distribution Constraints for Scalable and Efficient Image Retrieval

**Mohammad Omama, Po-han Li, Sandeep Chinchali**
The University of Texas at Austin
{mohd.omama,pohanli,sandeepc}@utexas.edu

## Abstract

Image retrieval is a crucial problem in robotics and computer vision, with downstream applications in robot place recognition and vision-based product recommendations. Modern retrieval systems face two key challenges: scalability and efficiency. State-of-the-art image retrieval systems train specific neural networks for each dataset, an approach that lacks scalability. Furthermore, since retrieval speed is directly proportional to embedding size, existing systems that use large embeddings lack efficiency. To tackle scalability, recent works propose using off-the-shelf foundation models. However, these models, though applicable across datasets, fall short in achieving performance comparable to that of dataset-specific models. Our key observation is that, while foundation models capture necessary subtleties for effective retrieval, the underlying distribution of their embedding space can negatively impact cosine similarity searches. We introduce Autoencoders with Strong Variance Constraints (AE-SVC), which, when used for projection, significantly improves the performance of foundation models. We provide an in-depth theoretical analysis of AE-SVC. Addressing efficiency, we introduce Single-shot Similarity Space Distillation ((SS)$_2$D), a novel approach to learn embeddings with adaptive sizes that offers a better trade-off between size and performance. We conducted extensive experiments on four retrieval datasets, including Stanford Online Products (SoP) and Pittsburgh30k, using four different off-the-shelf foundation models, including DinoV2 and CLIP. AE-SVC demonstrates up to a $16\%$ improvement in retrieval performance, while (SS)$_2$D shows a further $10\%$ improvement for smaller embedding sizes.

## 1 Introduction

Image retrieval involves finding the closest match of a given image (query) in a vast database of images (often called the reference set). It has numerous applications, from product recommendations in e-commerce to place recognition in robotics. In general, state-of-the-art (SOTA) image retrieval systems are dataset-specific. They are trained in a contrastive manner on a training split of the data, typically hand-labeled for positive and negative pairs for each query. Imagine a scenario where a server has a vast reference database of fashion clothing images and receives query images from users to find the closest matches in the reference set. Creating a separate split and hand-labeling positive and negative pairs to train a dataset-specific model is infeasible. A more scalable solution is to use off-the-shelf feature extractors like Dino (Caron et al., 2021), DinoV2 (Oquab et al., 2024), CLIP (Radford et al., 2021), etc. However, the performance of off-the-shelf feature extractors is generally lower compared to dataset-specific models. **Q1 (Scalability): How can we enhance the performance of these off-the-shelf models in a completely unsupervised way?**

Another key problem in retrieval is that the retrieval speed is directly proportional to the size of the embeddings. For an embedding of size $d$ and a reference set with $N$ image vectors, the retrieval speed for a single query is $O(d \times N)$. This retrieval time becomes significant as the size of the reference set ($N$) increases, with modern retrieval systems containing millions of images. Additionally, the embedding size directly affects the storage space required for the reference set and the communication bandwidth needed to send the query vector to the server in distributed image retrieval systems, which are common in product recommendations and robotic place recognition. Standard dimensionality reduction techniques, such as Principal Component Analysis (PCA), aim to preserve

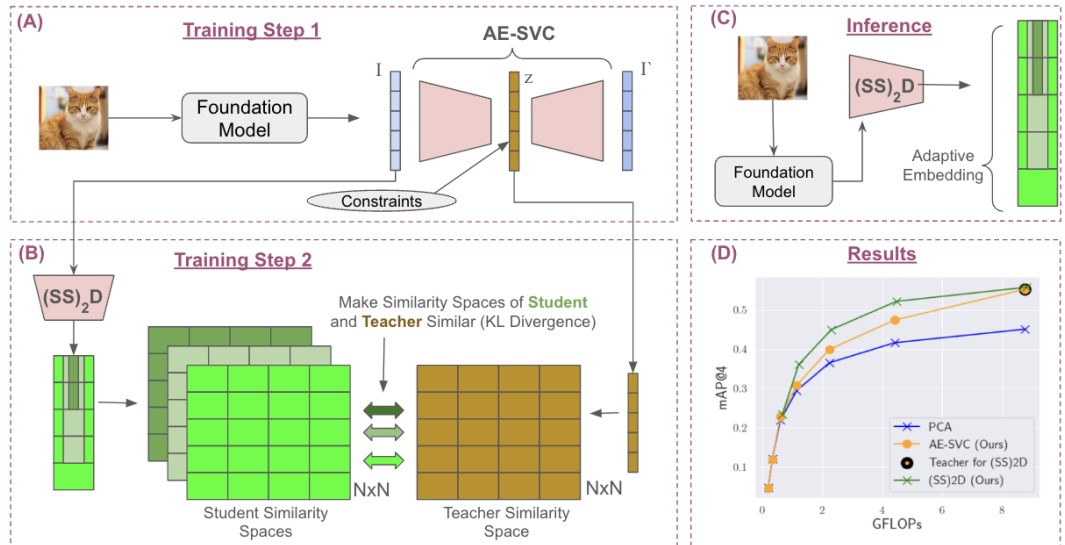

Figure 1: **Two-step pipeline for the proposed approach. (A)** AE-SVC (discussed in Sec. 3.1) trains an autoencoder with our constraints to improve foundation model embeddings. **(B)** (SS)₂D (discussed in Sec. 3.2) uses the improved embeddings from AE-SVC to learn adaptive embeddings for improved retrieval at any embedding size. **(C)** Once trained, (SS)₂D can be directly applied to foundation model embeddings to generate adaptive embeddings for improved retrieval. **(D)** AE-SVC (orange) boosts performance significantly, while (SS)₂D (green) further enhances results with smaller embeddings. Dino (blue) achieves optimal performance at 9 GLOPs, whereas (SS)₂D on top of AE-SVC achieves similar performance at only 2.5 GLOPs.

information, while Autoencoders (AEs) (Hinton & Salakhutdinov, 2006) and Variational Autoencoders (VAEs) (Kingma & Welling, 2013) focus on reconstruction quality. However, these methods are not optimized for retrieval tasks. Additionally, learning-based approaches like AEs and VAEs require separate training for each embedding size, which is impractical given varying compute, space, and communication constraints. **Q2 (Efficiency): Is there an effective unsupervised dimensionality reduction method that strongly preserves the similarity structure of the full embeddings, and is adaptive, i.e., does not need to be trained for each dimension separately?**

**Contributions:** To address **Q1 (Scalability)**, we propose Autoencoders with Strong Variance Constraints (AE-SVC). Our primary idea is that, while foundation models capture the necessary details for effective retrieval, the underlying distribution of their embeddings can negatively impact cosine similarity searches. AE-SVC trains an autoencoder while enforcing three constraints on the latent space: an orthogonality constraint, a mean centering constraint, and a unit variance constraint. We empirically show and mathematically prove that these constraints cause a shift in the cosine similarity distribution, making it more discriminative. These constraints not only lead to more effective dimensionality reduction but also **outperform the complete embeddings of foundation models**. We discuss the motivation and technical details of AE-SVC in section 3.1. We also provide an in-depth mathematical analysis of AE-SVC in section 4. To address **Q2 (Efficiency)**, we propose Single Shot Similarity Space Distillation ((SS)₂D). (SS)₂D aims at reducing embeddings to smaller ones while preserving their similarity relationships. The embedding learned with (SS)₂D is adaptive, *i.e.*, smaller segments of the output embedding also perform well in retrieval tasks. In summary, our approach consists of two steps (as shown in Fig. 1): first, we use AE-SVC to enhance the baseline performance of foundation models. Then, we leverage the improved embeddings as a teacher for efficient dimensionality reduction through (SS)₂D.

Fig. 1(D) also shows the retrieval performance versus retrieval speed in giga-FLOPs (floating operations) on the Pittsburgh30K dataset (Torii et al., 2013) using the Dino model (Caron et al., 2021). Note that retrieval speed is a function of the dimension $d$ and the reference set size $N$, $O(d \times N)$, with $N = 17,000$ for Pittsburgh30K. The performance of PCA (blue) represents the default off-the-shelf Dino model. We can see that AE-SVC (orange) significantly enhances the performance of the off-the-shelf Dino model (blue). Notice that it outperforms the complete Dino embedding. Applying (SS)₂D on top of AE-SVC further improves retrieval performance at smaller embedding

sizes (green). Section 5 experimentally shows the advantages of the proposed approaches in detail on four different datasets using different foundation models.

## 2 RELATED WORK

**Deep Metric Learning:** Image retrieval is modeled as a Deep Metric Learning (DML) problem (Wang et al., 2017). DML focuses on learning latent representations where similar items are close and dissimilar items are far apart. Various techniques have been proposed (Oh Song et al., 2016; Schroff et al., 2015; Ustinova & Lempitsky, 2016; Sohn, 2016; Wang et al., 2019) to learn meaningful latent representations for effective retrieval, including contrastive loss (Hadsell et al., 2006) and triplet loss (Schroff et al., 2015). More recent works use advanced N-pair loss (Sohn, 2016) or multi-similarity loss (Wang et al., 2019). Analogous to image retrieval, Visual Place Recognition (VPR) is also modeled as a DML problem (Garg et al., 2021). Various training objectives have been proposed for VPR (Ge et al., 2020; Xiao et al., 2023; Leyva-Vallina et al., 2023; Berton et al., 2022; Shubodh et al., 2024) to learn latent representations for effective retrieval. However, all of these approaches are dataset-specific, meaning they train specific neural networks for each dataset.

**Foundation Models in Image Retrieval:** The rise of foundation models, such as DINOv2 (Oquab et al., 2024), CLIP (Radford et al., 2021), ViT (Dosovitskiy et al., 2021), has allowed recent works to explore the applicability of these off-the-shelf models in image retrieval. As shown in (Keetha et al., 2023; Lu et al., 2024), off-the-shelf models, in some cases, can have comparable retrieval performance to dataset-specific models. However, these approaches either use the off-the-shelf models as-is (Keetha et al., 2023; Omama et al., 2023; han Li et al., 2024) or adapt them using a labeled held-out dataset (Lu et al., 2024). In stark contrast, our approach, `AE-SVC`, exploits the inherent properties of the distribution to improve the performance of these off-the-shelf models without requiring any labeled data.

**Distribution Constraints in Semi-supervised Representation Learning:** Semi-supervised learning has seen significant advances with methods that leverage self-supervised objectives to learn robust representations. A notable approach is Barlow Twins, which employs an additional cross-correlation loss to align representations while minimizing redundancy across dimensions, promoting feature decorrelation (Zbontar et al., 2021). Extending this idea, VICReg introduces an additional variance constraint that ensures sufficient spread in the learned features, preventing collapse and further enhancing representation quality (Bardes et al., 2022). However, to date, no work has specifically explored the impact of such constraints in the context of autoencoders for retrieval tasks. In this paper, we are the first to address this gap, providing a comprehensive theoretical analysis of these constraints and their impact on retrieval.

**Similarity Preserving Knowledge Distillation:** Similarity preserving knowledge distillation (SPKD) focuses on transferring knowledge from a teacher model to a student model, similar to standard distillation (Hinton et al., 2015; Buciluă et al., 2006; Ba & Caruana, 2014; Zagoruyko & Komodakis, 2017; Zhang et al., 2019; Passalis & Tefas, 2018), while maintaining the structure of similarity of the data in the feature space. One of the pioneering works in this area is the Fit-Nets approach, which uses intermediate representations to guide the student model (Romero et al., 2014). The attention transfer method extends this by aligning the attention maps of the student and teacher models (Zagoruyko & Komodakis, 2017). More recent advancements include (Tung & Mori, 2019), which explicitly preserves pairwise similarities between data points in the feature space during distillation. Contrastive Representation Distillation (CRD) further improves performance by using contrastive loss to maximize mutual information between teacher and student representations (Tian et al., 2020). Additionally, the Relational Knowledge Distillation (RKD) approach emphasizes the importance of preserving the relational knowledge among data points (Park et al., 2019). All of these works focus on training a lightweight student network to mimic a heavy teacher network. We are the first to explore this idea in a dimensionality reduction setting for image retrieval.

**Adaptive Feature Embeddings:** A core requirement for dimensionality reduction in retrieval systems is adaptive embeddings due to varying compute constraints. Previous works on adaptive feature embeddings have created efficient and adaptive embedding spaces for compression (Li et al., 2023) and image classification (Kusupati et al., 2022), while ours focuses on image retrieval. The key idea is to use only one neural network model to output a feature embedding ensuring that smaller sub-chunks of the embedding also perform well in retrieval tasks.

## 3 METHODOLOGY

We now formally define the image retrieval problem:

Suppose we have a query set $Q$ of $M$ query image feature vectors and a reference set $R$ of $N$ reference image feature vectors, where each vector is $d$-dimensional. Given $q_j \in \mathbb{R}^d$ randomly drawn from $Q$, the task is to identify the feature vector $r_{i*}$ in $R$ that has the smallest distance or highest similarity with $q_j$. This can be mathematically expressed as:

$$r_{i*} = \arg \min_{r_i \in R} \mathrm{dis}(q_j, r_i),$$

with the distance dis generally based on cosine similarity, i.e., $\mathrm{dis}(q_j, r_i) = 1 - \cos(q_j, r_i)$, where cos denotes the cosine similarity.

We will now describe our approach in detail. First, we will explain the step-by-step procedure for `AE-SVC`. Next, we will present an in-depth explanation of $(SS)_2D$. Note that both `AE-SVC` and $(SS)_2D$ utilize only the reference set $R$ during the training time. The query set $Q$ is assumed to be unavailable to the user during training. A standard assumption when doing dimensionality reduction in image retrieval settings (Keetha et al., 2023).

### 3.1 AUTOENCODERS WITH STRONG VARIANCE CONSTRAINTS (AE-SVC)

**Motivation:** Our primary motivation for `AE-SVC` is that, while foundation models capture necessary subtleties for effective retrieval, the underlying distribution of their embedding space can negatively impact cosine similarity searches. In Sec. 4, we discuss in mathematical detail that the discriminative power of the retrieval task is maximized when the variance of the cosine similarity distribution is minimized. `AE-SVC` introduces three constraints on the latent space: an orthogonality constraint, a mean centering constraint, and a unit variance constraint. In Sec. 4, we prove that these constraints on the latent space minimize the variance of the cosine similarity distribution. We now provide a more intuitive explanation of why these constraints are helpful. Consider the example illustrated in Fig. 2. We have a reference set comprising four categories of clothing: men's tank-tops, men's half-shirts, women's tank-tops, and women's half-shirts. The task is to match

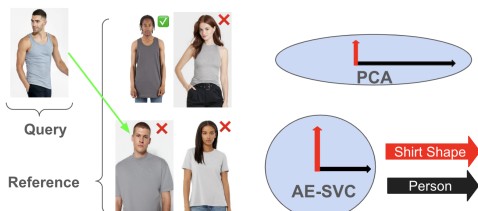

Figure 2: **In standard dimensionality reduction (say PCA), cosine similarity is disproportionately influenced by high-variance dimensions, leading to poor retrieval.** Given a task to match the query with the correct clothing type. A query image of a white man in a tank-top may be incorrectly matched to a white man in a half-shirt due to the dominant person dimension. Ideally, both orthogonal dimensions should have an equal influence on cosine similarity.

the query with the correct clothing type. After applying dimensionality reduction (like a standard PCA) to the reference set, we obtain two orthogonal dimensions: a person dimension (with high variance) and a shirt shape dimension (with low variance). If similarity is overly dependent on the high-variance dimension, as shown mathematically in Sec. 4, retrieval performance suffers. For instance, a query for a white man in a tank-top might incorrectly match with a white man in a half-shirt due to the dominant person dimension. Ideally, we need a space where both the person and shirt dimensions equally influence the distinction between men and women, as well as between half-shirts and tank-tops. The goal of `AE-SVC` is to learn a projection from the reference set $R$ such that the latent dimensions are orthogonal and have equal variance.

**Approach:.** `AE-SVC` (as shown in Fig. 1(**A**)) takes an input embedding $I \in \mathbb{R}^d$ coming from the foundation model, encodes it to a latent representation $z \in \mathbb{R}^d$, and produces a reconstruction $I' \in \mathbb{R}^d$. We have three additional losses to enforce the constraints mentioned above: an orthogonality constraint, a mean centering constraint, and a unit variance constraint. Note that we can choose any size for the latent representation $z$. We show the results for multiple sizes of $z$. However, since `AE-SVC` is used to guide $(SS)_2D$ later, we will use the best performing embedding, i.e., $z \in \mathbb{R}^d$, for this guidance.

**Reconstruction Loss:** We minimize the Mean Squared Error (MSE) between the input embedding $I$ and the reconstruction $I'$:

$$\mathcal{L}_{\text{rec}} = \frac{1}{n} \sum_{i=1}^{n} \|I_i - I'_i\|_2^2, \tag{1}$$

where $n$ is the number of samples.

**Covariance Loss:** To promote orthogonality, we penalize off-diagonal terms of the covariance matrix:

$$\mathcal{L}_{\text{cov}} = \left\| \frac{1}{n}(Z - \mu)^\top (Z - \mu) - \mathbb{I} \right\|_F^2, \tag{2}$$

where $Z \in \mathbb{R}^{n \times d}$ is the matrix of latent representations, $\mu \in \mathbb{R}^d$ is the mean of $Z$, and $\mathbb{I}$ is the identity matrix.

**Variance Loss:** We constrain the variance of each latent dimension to be one:

$$\mathcal{L}_{\text{var}} = \frac{1}{d} \sum_{i=1}^{d} \left( \text{Var}(z^i) - 1 \right)^2, \tag{3}$$

where $z^i$ is the $i$-th latent dimension of $z$ and $\text{Var}(z^i)$ is the variance of the $i$-th latent dimension across all $n$ samples.

**Mean-Centering Loss:** To ensure the latent space is centered around zero, we minimize the mean of the latent dimensions:

$$\mathcal{L}_{\text{mean}} = \frac{1}{d} \sum_{i=1}^{d} (\mu^i)^2, \tag{4}$$

where $\mu^i$ is the mean of the $i$-th latent dimension across all $n$ samples.

**Total Loss:** The final objective combines all four losses:

$$\mathcal{L}_{\text{AE-SVC}} = \lambda_{\text{rec}} \mathcal{L}_{\text{rec}} + \lambda_{\text{cov}} \mathcal{L}_{\text{cov}} + \lambda_{\text{var}} \mathcal{L}_{\text{var}} + \lambda_{\text{mean}} \mathcal{L}_{\text{mean}} \tag{5}$$

where $\lambda_{\text{rec}}, \lambda_{\text{cov}}, \lambda_{\text{var}}$, and $\lambda_{\text{mean}}$ are hyperparameters discussed in A.6. By optimizing this loss, AE-SVC learns a latent space that improves retrieval performance through accurate reconstruction, decorrelated and normalized feature dimensions, and a zero-centered mean. The latent representation, $z \in \mathbb{R}^d$, is then used to guide the training of $\texttt{(SS)}_2\texttt{D}$.

## 3.2   SINGLE SHOT SIMILARITY SPACE DISTILLATION ($\texttt{(SS)}_2\texttt{D}$)

**Motivation:**  As previously mentioned, embedding size significantly impacts retrieval speed. For an embedding of size $d$ and a reference set with $N$ image vectors, the retrieval speed for a single query is $O(d \times N)$. Therefore, it is crucial to reduce the embedding size while maintaining performance. Standard dimensionality reduction techniques, such as PCA, Autoencoders (AEs), and Variational Autoencoders (VAEs), are not optimized for retrieval. Additionally, AEs and VAEs require separate training for each embedding size, which is impractical given varying computational constraints. Therefore, there is a need for adaptive embeddings that maintain retrieval performance and are flexible, ensuring that smaller sub-chunks of the embedding also perform well in retrieval tasks.

**Approach:** Previous works have employed Similarity-Preserving Knowledge Distillation (SPKD) both in hidden layer activations (Smith et al., 2023) and output layers (Wu et al., 2022). SPKD trains a student network to ensure that input pairs producing similar (or dissimilar) activations in the teacher network yield similar (or dissimilar) activations in the student network. However, these studies have not explored similarity-preserving distillation for efficient dimensionality reduction. Our approach is conceptually similar to SPKD, but rather than focusing on network distillation (teacher to student), we perform embedding-level distillation, reducing a large embedding to a smaller one while preserving similarity relationships. Using SPKD directly for dimensionality reduction requires training separate networks for each dimension size. Previous works on adaptive embeddings have utilized random projections in a distributed compression setting (Li et al., 2023) or optimized multi-class classification loss for each nested dimension (Kusupati et al., 2022). Following the approach in (Kusupati et al., 2022), we optimize a similarity-preserving loss for each nested dimension. By combining a similarity-preserving loss with an adaptive embedding training pipeline, we introduce our approach, Single Shot Similarity Space Distillation, or $\texttt{(SS)}_2\texttt{D}$. Fig. 1(**B**) illustrates the central idea of $\texttt{(SS)}_2\texttt{D}$ which we will now describe in detail.

Let $z \in \mathbb{R}^d$ be the latent representation of the reference set obtained via `AE-SVC`. Our goal is to learn a smaller embedding such that the performance of the smaller embedding approximates that of the full embedding. In essence, the complete embedding $z$ serves as a guide for creating smaller embeddings. Although `(SS)₂D` could, in theory, directly use the foundation model embedding $I$ for guidance, the latent representation $z$ is shown to perform better in retrieval tasks, thus providing stronger guidance.

Consider a set $\mathcal{M} \subset [d]$ of embedding sizes and a neural network $\mathcal{F}(\cdot; \theta)$ parameterized by $\theta$. We want to learn a projection $\hat{z} = \mathcal{F}(I; \theta)$ such that each of the first $m$ dimensions, for $m \in \mathcal{M}$, of the learned embedding vector ($\hat{z}^{1:m} \in \mathbb{R}^m$) independently preserves the retrieval performance of $z$. To learn the function $\mathcal{F}$, we first compute a cosine similarity matrix $C \in \mathbb{R}^{N \times N}$ by calculating the cosine similarities of every $z$ with every other $z$ in the reference set $R$. We refer to this matrix as the teacher's cosine similarity space. Let $C_i$ denote the $i^{\text{th}}$ row of $C$.

Next, for each $m \in \mathcal{M}$, we define a student similarity space $\tilde{C}^m \in \mathbb{R}^{N \times N}$ calculated using $\hat{z}^{1:m}$. We then introduce a Kullback-Leibler (KL) divergence loss $l^m$ between $\tilde{C}^m$ and $C$ for each $m$ as follows:

$$l^m = \sum_i D_{\text{KL}}(\tilde{C}_i^m \| C_i). \tag{6}$$

The overall loss $L$ is then expressed as:

$$L_{\text{(SS)}_2\text{D}} = \sum_m l^m = \sum_m \sum_i D_{\text{KL}}(\tilde{C}_i^m \| C_i). \tag{7}$$

## 4 THEORETICAL ANALYSIS OF `AE-SVC`

In Section 3.1, we stated that the discriminative power of the retrieval task is maximized when the variance of the cosine similarity distribution is minimized. Given a background distribution $B$ (in our case, the distribution of the reference set) and a distribution $S$ representing vectors of a specific class, we define the discriminative power using a statistical threshold $\tau(\alpha)$, where $\tau$ is the quantile of $B$ at confidence level $\alpha$. Specifically, the discriminative power is the probability that the cosine similarity between two vectors $S_a, S_b \in S$ exceeds this threshold:

$$P\left(\cos(S_a, S_b) > \tau(\alpha)\right) = \text{erf}\left(\frac{\mathbb{E}[\cos(S_a, S_b)] - \tau(\alpha)}{\sqrt{\text{Var}[\cos(S_a, S_b)]}}\right), \tag{8}$$

where $\text{erf}$ denotes the error function. Eq. 8 can be reformulated in terms of the variance of the background cosine similarity distribution, demonstrating that the discriminative power is maximized when this variance is minimized. For a detailed proof of this result, along with alternative approaches to modeling discriminative power, we refer the reader to Smith et al. (2023).

We now verify if the results of Gaussian assumptions also hold for real datasets. Fig. 3 shows the distribution of cosine similarities of the reference set feature vectors on the Pittsburgh30k dataset (Torii et al., 2013). We compute cosine similarities between all pairs of feature vectors in the reference set and plot the probability mass function (PMF) with and without `AE-SVC` for two models: the off-the-shelf foundation model Dino (Fig. 3a) and the SOTA dataset-specific model Cosplace (Berton et al., 2022) (Fig. 3b). Cosplace, which excels at retrieval, exhibits a PMF with significantly less variance compared to Dino, as seen in the blue curves in Fig. 3a and 3b. The lower variance indicates more discriminative feature vectors, which aligns with the findings from Eq. 8 that smaller variance enhances discriminative power and retrieval performance. Applying `AE-SVC` reduces the variance in both models (orange curves), with a more pronounced shift in Dino than in Cosplace. This suggests that `AE-SVC` benefits the off-the-shelf foundation model more ($10\%$) than the dataset-specific model ($2\%$), as reflected in the retrieval performance plot in Fig. 3c. Thus, the remainder of this work focuses on off-the-shelf foundation models.

We have mathematically and empirically demonstrated that the discriminative power of the retrieval task is maximized when the variance of the cosine similarity distribution is minimized. Next, we will

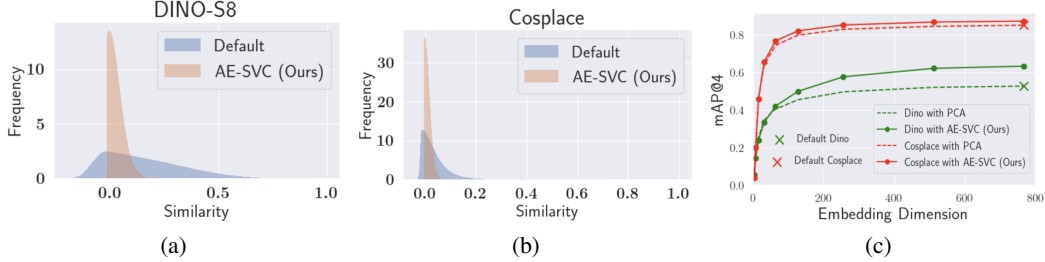

(a)

(b)

(c)

Figure 3: `AE-SVC` reduces the variance of cosine similarity distributions in both foundation (a) and dataset-specific models (b), with a more significant shift in foundation models (a). This results in greater improvement in retrieval performance for the foundation model (Dino) compared to the dataset-specific model (Cosplace), as shown in (c).

prove that the constraints introduced on the latent space in Sec. 3.1 indeed minimize this variance. Let $p(z)$ be the distribution of `AE-SVC`, i.e., $z \sim p(z)$. By applying constraints in Eq. 2 and Eq. 4, the $p(z)$ has a zero mean, $\mu = 0$, and a diagonal covariance matrix $\Sigma \in \mathbb{R}^{d \times d}$, with eigenvalues $\lambda_i = \sigma_i^2$, where $\sigma_i^2$ is the variance of the individual dimensions of $z$. Given two vectors $X$ and $Y$, $X, Y \sim p(z)$, the cosine similarity between $X$ and $Y$ is:

$$\cos(X, Y) = \frac{\sum_{i=1}^{d} X_i Y_i}{\sqrt{\sum_i X_i^2} \sqrt{\sum_i Y_i^2}} = \frac{X^T Y}{\sqrt{(X^T X)(Y^T Y)}} = \frac{X^T Y}{\|X\| \|Y\|}, \quad (9)$$

where subscript $i$ denotes the $i$-th element of a vector , and $\|X\|$ and $\|Y\|$ denote the Euclidean norms of vectors $X$ and $Y$ respectively. Each term in the numerator of Eq. 9 is just a product of two random variables, but the denominator makes things complicated. A closed-form solution of a ratio of two random variables is difficult to calculate.

Let us introduce a relaxation that can be used to approximate the denominator and simplify the calculation. Using cosine's scale invariance (shown in A.2), we define a relaxation $r(z)$ as:

$$r(z) = \frac{z}{m}, \quad (10)$$

where $m = \sqrt{\sum_i \sigma_i^2}$. Relaxation $r$ has two properties. First, the expected value of the squared norm after relaxation, denoted as $\mathbb{E}[\|r(z)\|^2]$, is equal to 1. Second, the ratio of variance to square of the mean is given by $\frac{\sum_k 2\sigma_k^2}{(\sum_k \sigma_k^2)^2}$, and it approaches 0 as the dimensionality $d$ tends to infinity under the condition that the contribution of any single covariance eigenvalue is sufficiently small. Smith et al. (2023) show that these properties can then be used to approximate the cosine similarity as:

$$\cos(X, Y) = \cos(r(X), r(Y)) = \frac{\sum_{i=1}^{d} r(X)_i r(Y)_i}{\|r(X)\| \|r(Y)\|} \approx \sum_{i=1}^{d} r(X)_i r(Y)_i. \quad (11)$$

With this approximation, the moments of the cosine similarity can be easily calculated:

$$\mathbb{E}[\cos(X, Y)] = 0, \quad (12)$$

$$\text{Var}(\cos(X, Y)) = \sum_{i=1}^{d} \frac{\sigma_i^4}{(\sum_{j=1}^{d} \sigma_j^2)^2}. \quad (13)$$

We can see from Eq. 13 that the cosine similarity distribution is highly dependent on the dimensions with high variance. Taking the gradient of Eq. 13 with respect to $\sigma_i^2$, we get:

$$\frac{\partial}{\partial \sigma_i^2} \text{Var}(\cos(X, Y)) = 2 \left( \sum_{j=1}^{d} \sigma_j^2 \right)^{-3} \left[ (\sum_{j=1}^{d} \sigma_j^2) \sigma_i^2 - (\sum_{j=1}^{d} \sigma_j^4) \right]. \quad (14)$$

We can see that the global minimum of the approximation for $\text{Var}(\cos(X, Y))$ is achieved when all individual variances of the original distribution are equal, i.e., $\sigma_i^2 = \sigma_j^2 \ \forall \ i, j$. This is enforced

in `AE-SVC` by the variance constraint (Eq. 3). Therefore, we theoretically and empirically justified that `AE-SVC` minimizes the variance of the cosine similarity distribution, improving retrieval performance.

# 5 EXPERIMENTS

## 5.1 EXPERIMENTAL SETUP

**Datasets**: We evaluate our approach on four distinct image retrieval datasets: InShop (Liu et al., 2016), Stanford Online Products (SOP) (Song et al., 2016), Pittsburgh30K (Torii et al., 2013), and TokyoVal (Torii et al., 2015). InShop and SOP are state-of-the-art (SOTA) image retrieval datasets commonly used in metric learning research, while Pittsburgh30K and TokyoVal are recognized as SOTA place recognition datasets, frequently used in robotics research.

**Foundation Models Evaluated**: To demonstrate the broad applicability of our approach, we conducted experiments using four different foundational models trained with a variety of techniques and pre-training datasets: CLIP (Radford et al., 2021), DINO (Caron et al., 2021), DINOv2 (Oquab et al., 2024), and ViT (Dosovitskiy et al., 2021). The CLIP model was pre-trained on the LAION-2B dataset (Community, 2023) using a contrastive learning approach. Both DINO and ViT models were pre-trained in a self-supervised manner on the ImageNet dataset (Russakovsky et al., 2015). The DINOv2 model was pre-trained in a self-supervised manner on the LVD-142M dataset (Oquab et al., 2023). For the DINO and DINOv2 models, we evaluated different model sizes: specifically, DINO-S8 with 22.7 million parameters and DINO-B16 with 300 million parameters, as well as DINOv2-Small with 22.1 million parameters and DINOv2-Large with 300 million parameters.

**Settings**: Our work presents two contributions: the introduction of a novel space, `AE-SVC`, and a new method for learning an adaptive embedding, $(SS)_2D$, on top of `AE-SVC`. Consequently, our experiments are divided into two parts. First, we demonstrate that `AE-SVC` significantly enhances the retrieval performance of foundation models across various datasets and model choices. Second, we show that $(SS)_2D$ facilitates more effective dimensionality reduction on top of `AE-SVC`, further improving performance at lower dimensions. We compare $(SS)_2D$ with the following baselines: a Variational Auto Encoder (VAE) and PCA. Additionally, we compare $(SS)_2D$ with using Similarity Space Distillation applied to each dimension separately, referred to as SSD. Note that SSD serves as a theoretical upper bound for $(SS)_2D$. We use mean Average Precision at $k$ (mAP@$k$), a standard metric for evaluating retrieval performance. We show additional results on the Recall@$k$ metric in Appendix A.5.

## 5.2 RESULTS

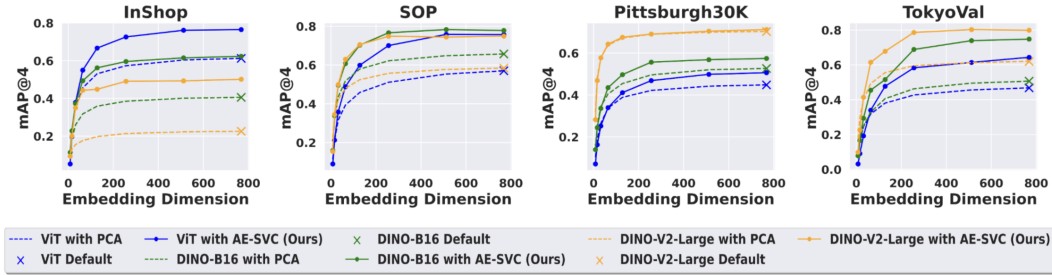

Figure 4: **`AE-SVC` significantly improves the retrieval performance of foundation models.** `AE-SVC` (solid lines) consistently outperforms the off-the-shelf foundation models, i.e., PCA (dashed lines), on four datasets, achieving a 15.5% average improvement in retrieval performance.

Fig. 4 shows the performance comparison between `AE-SVC` and PCA across 4 datasets, InShop, Stanford Online Products (SOP), Pittsburgh30K, and TokyoVal, using 3 foundation models (ViT, DINOv2-Large, and DINO-B16). The performance of PCA (dotted line) represents the default off-the-shelf performance of the foundation models, while the performance of `AE-SVC` is shown with solid lines. `AE-SVC` consistently outperforms PCA across all datasets and embeddings. For

example, the DINOv2-Large embedding (yellow) shows a $24\%$ improvement on InShop, $10\%$ on SoP, $2\%$ on Pittsburgh30K, and $22\%$ on TokyoVal at full embedding size. Overall, `AE-SVC` achieves an average improvement of $15.5\%$ across all datasets and embeddings at full size. Also, `AE-SVC` consistently surpasses PCA at smaller embedding sizes, providing a better size-performance trade-off. For additional results with more foundation models, see Appendix A.3.

Fig. 5 illustrates the advantages of $(SS)_2D$ on two datasets: InShop and TokyoVal, utilizing two foundation models, DINO-B16 and ViT. We compare $(SS)_2D$ with PCA (blue dashes) and a non-linear dimensionality reduction technique, Variational Auto Encoder (VAE) (black crosses). Additionally, we compare it with Similarity Space Distillation (SSD), which represents the theoretical upper bound of $(SS)_2D$. Note that unlike VAE (black crosses) and SSD (red crosses), which are trained for each dimension separately, $(SS)_2D$ learns an adaptive embedding in one shot. From the plot, we observe that `AE-SVC` (orange) initially enhances the performance of the original embedding (blue), and $(SS)_2D$ subsequently provides additional improvements of up to $10\%$ at smaller embedding sizes. Also note that since SSD is trained for each dimension separately, it serves as the theoretical upper bound of $(SS)_2D$. The results indicate that $(SS)_2D$ closely approaches this upper bound. For additional results involving more datasets and foundation models, please refer to Appendix A.4. As discussed in Sec. 1, a smaller embedding size directly translates to faster retrieval speeds. Therefore, the effective dimensionality reduction achieved with $(SS)_2D$ results in improved retrieval speeds, as demonstrated in Fig. 1(**D**).

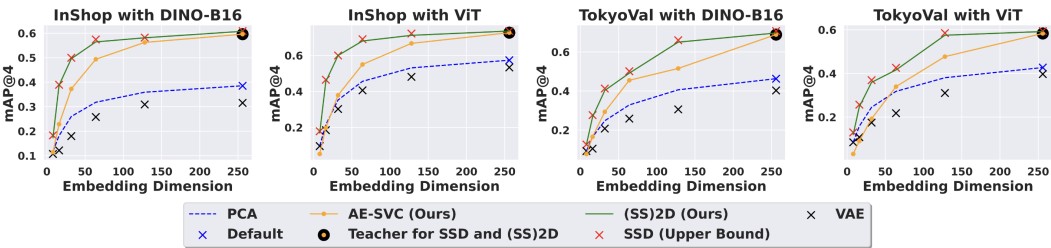

Figure 5: **Applying $(SS)_2D$ over `AE-SVC` leads to further performance boost at lower embedding sizes.** Compared to VAE and SSD, $(SS)_2D$ offers superior single-shot dimensionality reduction, achieving up to a $10\%$ enhancement at smaller embedding sizes, closely approaching SSD's theoretical upper bound.

## 5.3 LIMITATIONS

While `AE-SVC` demonstrates significant improvements when applied to foundation models, its ability to enhance dataset-specific models is limited. The pre-trained models tailored for a specific dataset already capture the nuances required for effective retrieval, leaving less room for further optimization via `AE-SVC`. Additionally, $(SS)_2D$ introduces a computational overhead during training due to the need to compute the loss functions for various embedding sizes. This overhead increases training time, though it is still substantially lower than the cost of retraining neural networks for different embedding sizes from scratch.

## 6 CONCLUSION AND FUTURE WORK

In this paper, we proposed `AE-SVC` and $(SS)_2D$, which improve retrieval scalability and efficiency. Our results show a significant improvement on $4$ datasets, from image retrieval to place recognition tasks. `AE-SVC` demonstrates up to a $15.5\%$ improvement in retrieval performance, and $(SS)_2D$ shows an improvement of $10\%$ for smaller embedding sizes, which further boosts retrieval efficiency. Future work includes new retrieval models that orchestrate the variance-aware property in training loss of foundation models. Although we observe the resulting difference in the cosine similarity distributions of the foundation models and the data set-specific models (Figure 3), the fundamental discrepancy between the embedding spaces remains unknown to the community.

REPRODUCIBILITY STATEMENT

In the theoretical analysis section (Sec. 4) of this work, we clearly state all the assumptions as and when necessary. Appendix A.1 and A.2 provides additional details necessary to understand the proofs in Sec. 4. We also discussed all the hyperparameters in Appendix A.6.

ACKNOWLEDGEMENT

This work was supported in part by National Science Foundation grants No. 2133481 and No. 2148186, and by the Office of Naval Research (ONR) under Grant No. N00014-22-1-2254. Any opinions, findings, conclusions, or recommendations expressed in this material are those of the authors and do not necessarily reflect the views of these agencies.

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

# A APPENDIX

## A.1 INVARIANCE OF ORTHOGONAL TRANSFORMATION

We show that cosine similarity is invariant to orthogonal transformation. Define $T_{\mathrm{orth}} \in \{T : T^\top T = \mathbf{I}\}$ as an orthogonal transformation. Following equation 9, we see that

$$
\begin{aligned}
\cos(T_{\mathrm{orth}}X, T_{\mathrm{orth}}Y) &= \frac{(T_{\mathrm{orth}}X)^\top T_{\mathrm{orth}}Y}{\|T_{\mathrm{orth}}X\|\|T_{\mathrm{orth}}Y\|} \\
&= \frac{X^\top \mathbf{I}Y}{\sqrt{X^\top \mathbf{I}X}\sqrt{Y^\top \mathbf{I}Y}} \\
&= \frac{X^\top Y}{\|X\|\|Y\|} \\
&= \cos(X, Y) \quad \square.
\end{aligned}
\tag{15}
$$

## A.2 COSINE SCALE INVARIANCE

Similar to the derivation in A.1, we now show that cosine similarity is invariant to scaling. Again, we define a constant $c$ and follow equation 9:

$$
\begin{aligned}
\cos\left(\frac{X}{c}, \frac{Y}{c}\right) &= \frac{X^\top Y/c^2}{\|X\|\|Y\|/c^2} \\
&= \frac{X^\top Y}{\|X\|\|Y\|} \\
&= \cos(X, Y) \quad \square.
\end{aligned}
\tag{16}
$$

## A.3 `AE-SVC` ADDITIONAL RESULTS WITH MORE FOUNDATION MODELS

In Fig. 4, we showed the advantage of using `AE-SVC` for embeddings of three models. Here, in Fig. 6, we show additional results with three more embeddings, namely: CLIP, DINO-V2-Small, and DINO-S8. We observe similar trends as previously noted.

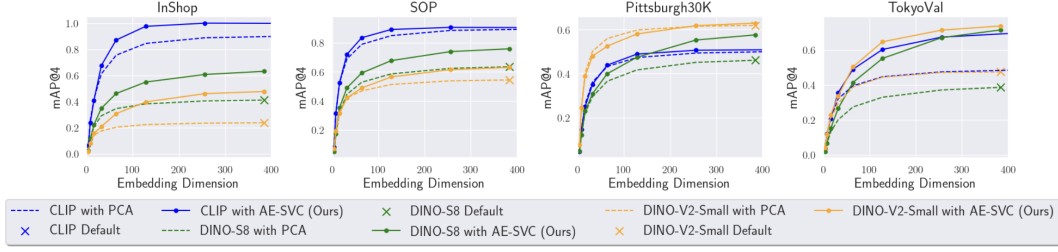

Figure 6: **`AE-SVC` significantly improves the retrieval performance of foundation models.** Here, we see a similar trend as discussed in the main paper but with three additional foundation models.

## A.4 (SS)$_2$D ADDITIONAL RESULTS WITH MORE DATASETS AND FOUNDATION MODELS

In Fig. 5, we showed the advantage of using (SS)$_2$D on top of `AE-SVC` for additional improvement in retrieval performance at smaller dimensions. Here, in Fig. 7, we show additional results for (SS)$_2$D on more datasets and embedding combinations. We observe similar trends as previously noted.

## A.5 `AE-SVC` ADDITIONAL RESULTS ON THE RECALL METRIC

In Fig. 4, we show the advantage of using `AE-SVC` for embeddings of three models on the mean Average Precision at $\mathbf{k}$ (mAP@$\mathbf{k}$) metric. Here, in Fig. 8, we show additional results on the Recall@$\mathbf{k}$ metric. We observe very similar trends on the Recall@$\mathbf{k}$ metric as we did on the mAP@$\mathbf{k}$ metric.

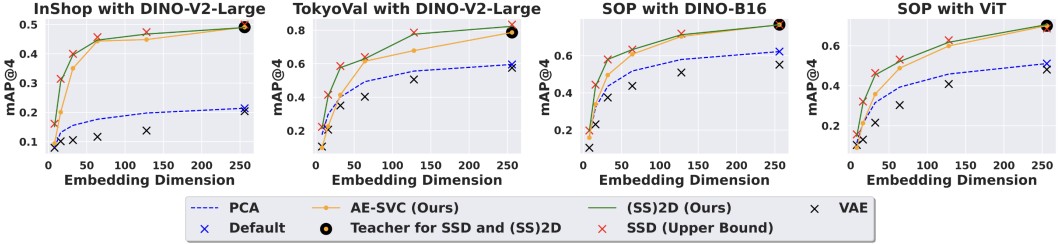

Figure 7: **Applying `(SS)`$_2$`D` over `AE-SVC` leads to further performance boost at lower embedding sizes.** Here, we see a similar trend as discussed in the main paper but with additional foundation models and datasets.

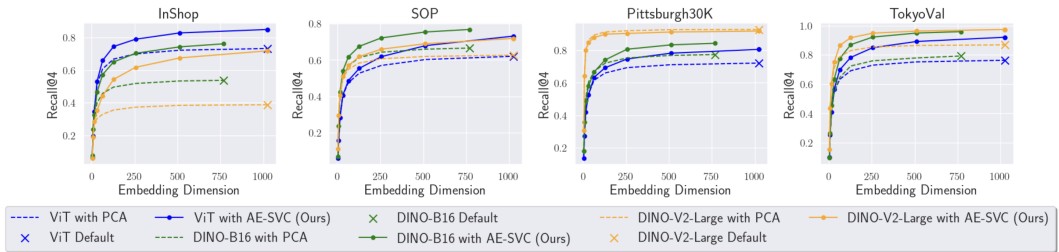

Figure 8: **`AE-SVC` significantly improves the retrieval performance of foundation models on the recall metric.** `AE-SVC` (solid lines) consistently outperforms the off-the-shelf foundation models, i.e., PCA (dashed lines), on four datasets.

## A.6 HYPERPARAMETERS

We discussed four hyperparameters in Sec. 3.1: $\lambda_{\text{rec}}$, $\lambda_{\text{cov}}$, $\lambda_{\text{var}}$, and $\lambda_{\text{mean}}$. We empirically determined the values of these hyperparameters, ensuring optimal performance across all datasets. The fixed values are: $\lambda_{\text{rec}} = 25$, $\lambda_{\text{cov}} = 1$, $\lambda_{\text{var}} = 15$, and $\lambda_{\text{mean}} = 1$.

## A.7 NETWORK SIZE ABLATIONS

Table 1: Retrieval performance (*map@4*) with different sizes of `AE-SVC` models at varying embedding dimensions. As the network size increases, retrieval performance scales up. However, the results show diminishing returns and possible overfitting with deeper networks.

| Model | 8 | 32 | 64 | 128 | 256 | 384 |
|---|---|---|---|---|---|---|
| $\text{AE-SVC}_1$ | 0.04 | 0.25 | 0.332 | 0.388 | 0.403 | 0.413 |
| $\text{AE-SVC}_2$ | **0.093** | 0.340 | 0.450 | 0.510 | 0.520 | 0.550 |
| $\text{AE-SVC}_3$ | **0.093** | **0.342** | **0.461** | **0.538** | **0.556** | **0.568** |
| $\text{AE-SVC}_5$ | **0.093** | 0.340 | 0.400 | 0.450 | 0.480 | 0.432 |
| $\text{AE-SVC}_{10}$ | 0.01 | 0.06 | 0.09 | 0.10 | 0.17 | 0.20 |
| PCA | 0.04 | 0.250 | 0.324 | 0.370 | 0.403 | 0.411 |

Table 1 shows the retrieval performance (*map@4*) of `AE-SVC` models with varying network sizes (indicated by the subscript $n$ in $\text{AE-SVC}_n$, where $n$ is the number of encoder/decoder layers) at different embedding dimensions (8, 32, 64, 128, 256, and 384). These experiments are done on the InShop dataset using the DINO-S8 feature extractor.

Larger *map@4* values ($\uparrow$) indicate better retrieval performance. The performance of linear `AE-SVC` ($\text{AE-SVC}_1$) is comparable to PCA. As the network size increases, retrieval performance scales up, with $\text{AE-SVC}_3$ achieving the best results across all dimensions. We have used $\text{AE-SVC}_3$ in all our experiments across datasets and embedding types.

However, performance begins to drop at $\texttt{AE-SVC}_5$ and significantly deteriorates at $\texttt{AE-SVC}_{10}$, indicating diminishing returns and possible overfitting with deeper networks. This behavior is consistent across all embedding dimensions.

## A.8 CONSTRAINT LOSSES APPLIED AT DIFFERENT LAYERS

Table 2 presents the retrieval performance (*map@4*) of various $\texttt{AE-SVC}$ configurations across different embedding dimensions (8, 32, 64, 128, 256, and 384). This evaluation uses a 3-layer encoder-decoder architecture. The constraint losses, which consist of variance, covariance, and mean-centering losses, are applied at different stages of the network. $\texttt{AE-SVC}$ applies these losses immediately after the encoder at the representation level. $\texttt{AE-SVC}^1$, $\texttt{AE-SVC}^2$, and $\texttt{AE-SVC}^3$ apply these losses after the first, second, and third decoder layers, respectively. Since the network has a 3-layer encoder-decoder architecture, $\texttt{AE-SVC}^3$ corresponds to applying these losses after the final decoder layer which aligns with existing representation learning methods such as Zbontar et al. (2021), Chen et al. (2020), and Kalantidis et al. (2021). In all configurations, a reconstruction loss is applied after the final decoder layer.

The proposed configuration ($\texttt{AE-SVC}$) consistently achieves the best retrieval performance across all embedding dimensions, highlighting the effectiveness of applying the loss at the representation level rather than later in the decoder network (like in previous approaches). Performance generally deteriorates when the loss is applied at deeper decoder layers, as observed in the results for $\texttt{AE-SVC}^1$, $\texttt{AE-SVC}^2$, and $\texttt{AE-SVC}^3$.

Table 2: Retrieval performance (*map@4*) with constraints applied at different layer of the $\texttt{AE-SVC}$ decoder. Consistent with our theoretical analysis in the main paper, the constraints should be applied directly after the encoder and we see a performance drop as they are applied later in the decoder.

| Model | 8 | 32 | 64 | 128 | 256 | 384 |
|---|---|---|---|---|---|---|
| $\texttt{AE-SVC}$ | **0.093** | **0.342** | **0.461** | **0.538** | **0.556** | **0.568** |
| $\texttt{AE-SVC}^1$ | 0.090 | **0.342** | 0.444 | 0.518 | 0.538 | 0.550 |
| $\texttt{AE-SVC}^2$ | 0.010 | 0.340 | 0.436 | 0.473 | 0.480 | 0.484 |
| $\texttt{AE-SVC}^3$ | 0.090 | 0.330 | 0.404 | 0.443 | 0.463 | 0.471 |

## A.9 ADDITIONAL RESULTS WITH MORE BASELINES AND HELD-OUT TRAIN SET

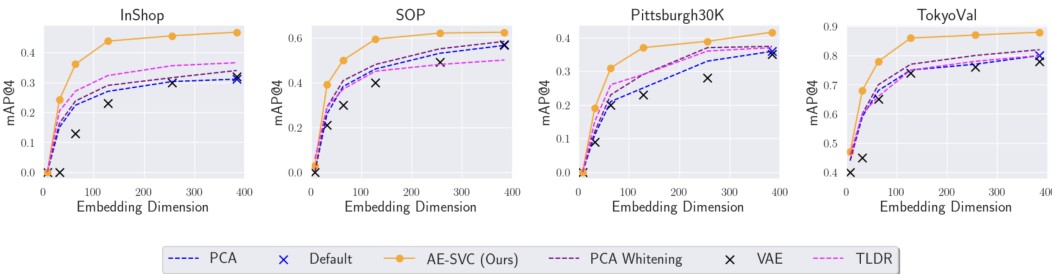

Figure 9: We show the retrieval performance of $\texttt{AE-SVC}$ (orange) compared against PCA (blue), PCA Whitening (purple), TLDR (magenta) and Variational Autoencoder (black). For this experiment, we use the Dino-S8 feature extractor. While PCA Whitening and TLDR improve the performance of the default feature extractor, $\texttt{AE-SVC}$ achieves a significantly greater improvement.

We have incorporated PCA Whitening (Hyvärinen et al., 2009) as a baseline for our approach ($\texttt{AE-SVC}$). We have also incorporated TLDR (Kalantidis et al., 2021), a recent dimensionality reduction work focussed on image retrieval. TLDR also focuses on dimensionality reduction for retrieval. However, TLDR uses only covariance loss. In stark contrast, we use a combination of three losses, including variance loss, which, as per our theoretical analysis, is the most important. Furthermore, following Barlow Twins, TLDR applies the covariance loss at the output of the decoder, whereas we apply it at the representation level.

