# OpenReview forum: "Exploiting Distribution Constraints for Scalable and Efficient Image Retrieval"
_ICLR.cc/2025/Conference — ICLR 2025 Poster_

### Official Review · Reviewer_Mq3y · 2024-10-30

**Soundness:** 2
**Presentation:** 3
**Contribution:** 2
**Rating:** 6
**Confidence:** 5

**Summary:**

This work uses a foundation model for image retrieval, making two contributions: (1) AE-SVC trains an autoencoder on top of the foundation model such that the latent vectors have statistical (mean and covariance) properties inspired or equivalent to centering, PCA and whitening. (2) SS2D performs dimensionality reduction jointly for multiple target dimensions by distilling a similarity matrix from the AE-SVC encoder to another student encoder. Both are learned on the test (reference) set, while the foundation model is pre-trained and frozen.

**Strengths:**

The idea is well motivated. The methodology is sound. The writing is of high quality and mostly clear. The experiments support the claims on a number of datasets and models. The two contributions are shown to be effective comparing with simple baselines.

**Weaknesses:**

The novelty is incremental, while important related work and baselines are missing. The setting of learning on the test set is questionable. Comparisons with baselines are lacking, not fair and not informative enough. Section 4 fails to properly separate existing work by Smith et al. (2023) from this work and is partly unclear. More details follow.

1. The paper addresses image retrieval and performs evaluation on standard deep metric learning and place recognition datasets. It misses revelant foundational work on image retrieval, including methodologies, benchmarks and protocols:

    > [*1] Gordo et al. 2016, Deep Image Retrieval: Learning Global Representations for Image Search
    >
    > [*2] Radenovic et al. 2017, Fine-Tuning CNN Image Retrieval with No Human Annotation
    >
    > [*3] Radenovic et al. 2018, Revisiting Oxford and Paris: Large-Scale Image Retrieval Benchmarking
    >
    > [*4] Weyand et al. 2020, Google Landmarks Dataset v2: A Large-Scale Benchmark for Instance-Level Recognition and Retrieval

    This work should be repositioned against this work in terms of novelty and protocols used. The related work and experiments sections should be extended accordingly, considering additional training and test datasets and evaluation protocols from these works.

2. The motivation for AE-SVC is scalability. That is, it is challenging to have a labeled dataset-specific training set, hence it is suggested to learn in an unsupervised on the test (reference) set. On one hand, this setting is extremely problematic because it transfers the burden from training to testing, by having to learn something on the test set. The latter is in many practical applications constantly changing, hence the model would need to be constantly updated. It is a kind of overfitting to the test set that has never been used in standard work like the references above. To my knowledge, it has only been used in

    > [*5] Liu et al. 2019, Guided Similarity Separation for Image Retrieval

    Hence, this work should be a baseline for comparison if the authors insist in their setting of learning on the test set. On the other hand, if the motivation to learn in an unsupervised way, this can still be done on a separate training set. For example:

    > [*6] Kim et al. 2022, Self-Taught Metric Learning without Labels

    Besides, learning in [2*] may be supervised, but with labels obtained automatically without human annotation. Self-supervised learning should not be a reason to learn on the test set.

3. In AE-SVC, I believe variance loss (3) is redundant. It is merely the diagonal part of covariance loss (2). Other than that, the covariance loss (2) and centering loss (4) are inspired by centering, PCA and whitening. Hence, AE-SVC should be compared to a baseline where centering, PCA and whitening are performed as post-processing on features extracted by the foundation model, without learning. This baseline is missing. Whitening has a long history in image retrieval but ignored here. Also very successful is supervised discriminative whitening [*2], which could be adapted for self-supervision to be used as a baseline.

4. In SS2D, I believe the loss is the same as

    > [*7] Park et al. 2019, Relational Knowledge Distillation

   The authors cite this paper but do not discuss the actual relation with their work. I believe the only new component here is applying it for dimensionality reduction and learning for multiple target dimensions at once, which are pretty incremental contributions. Also very relevant is

    > [*8] Kalantidis et al. 2022, TLDR: Twin Learning for Dimensionality Reduction

    in the sense that it is also learning a kind of autoencoder on top of frozen feature for dimensionality reduction. Instead of a reconstruction loss, it uses the Barlow Twins loss on the output vectors, which is the same as the covariance loss here, but uses cross-correlation instead. Both the architecture and cross-correlation matrix should be compared to as baselines.

5. Section 4 is useful and interesting, but it is presented as new to a large extent while it actually all comes from Smith et al. (2023). The only new part is the experiment of Figure 3. In addition, parts are copied from Smith et al. (2023) without carrying over all relevant information, such that the resulting text is unclear. For example, (8) refers to "discriminative power" but there is nothing to discriminate here, whereas Smith et al. (2023) compare the distribution of target similarities to a "null" or "background" distribution where the term "discriminative" makes sense. As another example, (14) results in a "global minimum" without saying anything about the function being convex, which is shown indeed by Smith et al. (2023).

    In the analysis of Section 4, the data is taken to be centered and nothing is said about the effect of this choice to the distribution of similarities, either here or in Smith et al. (2023). To my understanding, the variance of similarities is a **concave** function of the mean and is **maximized** when the mean is zero. This is the opposite of what happens with whitening and raises questions on the motivation of minimizing the variance. To the very least, the experiment of Figure 3 should be extended to also show the distribution of similarities with centered data but without PCA and whitening. It should also include centering, PCA and whitening as post-processing steps rather than being learned with (2), (3) and (4).

6. In the experiments of Figures 4 and 5, the current work (AE-SVC, SS2D or SSD) uses an additional encoder on top of the foundation model, while the "default" and "PCA" are using the foundation model alone, hence are not comparable. These results tell us nothing about whether the benefit is coming from the properties encouraged by the loss functions, the additional learned parameters or the adaptation to the test set. These are three factors that should be evaluated independently in some way, for example by fine-tuning the last layers of the foundation model rather than adding an extra encoder.

    Whitening by post-processing after PCA should be definitely added as a baseline. TLDR or other state of the art dimensionality reduction methods should also be added, because centering, PCA, whitening and VAE are very basic baselines.

## Post-rebuttal

The following is an assessment after rebuttal and considering the other reviews.

The contribution is incremental, which is not necessarily a problem if there is a clear and fair positioning with respect to previous work, a clear message and good results. The situation has clearly improved during rebuttal and discussion, but I am afraid that the paper is still weak in these three aspects. In particular:

- *Losses*. The authors state in the dicussion that there is "stark contrast" with TLDR in that they use three losses rather than one. The covariance loss is very close to Barlow Twins loss used in TLDR, which is cross-correlation. An experimental comparison between the two is not there. The variance loss only refers to the diagonal of the covariance loss and treating it separately makes sense only because of using a different loss weight, which is not mentioned anywhere in the main text. The appendix A6 says $\lambda_{\text{cov}}=1$, $\lambda_{\text{var}}=15$, but the effect of each $\lambda$ is not shown. What if $\lambda_{\text{cov}}=15$ for example? A contribution is meaningful when effective, especially if incremental. The mean-centering loss is something ubiquitous. The relative benefit of each loss component is not shown anywhere. Under these conditions, the contribution claim of three losses is still not adequately supported by evidence.

- *Theoretical support*. The authors claim in the paper and discussion that providing theoretical analysis in section 4 is a contribution but this study is carried out by Smith at al. (2023) and section 4 does not clarify if there is anything new here except the experiment of Figure 3. This is kind of over-claiming contribution. Minimizing the variance of cosine similarity distribution is still not a clear narrative as motivation since, as a function of the mean, it is maximized. Using the term "discriminative" when there is nothing to discriminate from is still confusing.

- *Distillation*. The authors state in the paper that their approach "is conceptually similar to SPKD", but this is different from saying "we use RKD (Park et al., 2019)", for example. Again, this is kind of over-claiming contribution.

- *Supervision*. On one hand, there is a supervised PCA whitening approach that is common in image retrieval, as performed by Radenovic et al. (2017) mentioned in my review. This would be another baseline, plus the authors could use this kind of supervision on their own approach. On the other hand, there are more unsupervised metric learning methods like Kim et al. (2022) that I have mentioned in my review, which could be used for distillation or self-supervision, as competitors or as components of this method.

- *Two-step*. The authors state in the discussion that their approach is two-step in "stark contrast" with previous work, but they do not experiment with an one-step variant of their approach to show the benefit and they do not explain why two-step is better.

The most important additions during the rebuttal are the experiment of training on a held-out training set, the comparison with TLDR and applying the losses at different layers (Table 2 of rebuttal document). The last one possibly accounts partially for the performance improvement over TLDR but the effect of the losses is not shown. However, as discussed above, several things are still missing in terms of fair positioning with previous work, some rewriting to make the message clearer and more experiments. I believe all of these issues could be resolved in theory, in which case the paper would be interesting enough to publish despite the incremental contribution. In case the paper is accepted, I strongly recommend the authors to revise accordingly for the camera-ready.

**Questions:**

1. The mean and covariance quantities in (4) and (2) (also variance in (3), but this is redundant as discussed above), are global in the sense that they are measured on all $n$ samples of the test set. What happens when the test set is large and how are they treated when learning with a stochastic optimizer in mini-batches? Are mini-batches used in this work?

2. Can the authors confirm or refute my understanding that the variance of similarities is a concave function of the mean and is maximized when the mean is zero? In the latter case, how would they update the paper?

---

### Official Review · Reviewer_vMAk · 2024-10-31

**Soundness:** 2
**Presentation:** 3
**Contribution:** 2
**Rating:** 6
**Confidence:** 3

**Summary:**

This paper aims to improve image retrieval under the context of using pretrained image foundation models. Two concrete techniques are designed: (1) Autoencoders with Strong Variance Constraints (AE-SVC) regularizes the embeddings from foundation models to improve the the retrieval performance, (2) Single-shot Similarity Space Distillation ((SS)_2D) compresses the embedding in an adaptive way to accelerate the retrieval. The paper provides theoretical and empirical analyses to justify the effectiveness of AE-SVC. The experimental results show that the proposed approaches can effectively improve the retrieval performance/efficiency of embedding from four different visual foundation models across four datasets.

**Strengths:**

1. The proposed approaches are quite simple yet effective. AE-SVC is backed up by some reasonable empirical analyses in Sec.4. The adaptive dimension reduction brought by (SS)$_2$D is interesting and can be practically useful to avoid training multimodal models for different embedding dimensions.

2. The authors conduct thorough experiments to demonstrate the effectiveness of the proposed approaches: 4 image retrieval datasets and 4 visual foundation models are considered, and AE-SVC + (SS)$_2$D consistently show positive results for efficient and accurate image retrieval.

3. A codebase is provided in the supplementary material. Roughly looking through the code, all necessary model/data components are included, ensuring the reproducibility of the paper.

**Weaknesses:**

1. The theoretical analysis in Sec.4 is only conducted based on the Gaussian distribution assumption, which does not necessarily hold for the embeddings from the visual foundation models. Based on the PDF visualization in Figure 3, the true distribution of cosine similarity between image embeddings is far from Gaussian.

2. In large-scale retrieval, approximate nearest neighbour (ANN) is commonly used to accelerate the retrieval process, so the complexity analysis of image retrieval in the paper is not necessarily right in real-world applications.

3. The figure placement can be optimized. The method details should not be in the first figure, which increases the burden of understanding the introduction, and forces people to switch back and forth when reading the method section.

**Questions:**

The paper focuses on improving the embedding from image foundation models by unsupervised training without any labels. However, given the recent advancements in multimodal LLMs, it seems possible to leverage these models (like GPT-4o) to automatically produce reasonable labels for fine-tuning image embedding models. If this is feasible, would the proposed approaches still be practically useful, given the fact mentioned in the paper that dataset-specific models are better than foundation models?

---

### Official Review · Reviewer_f9Kc · 2024-11-01

**Soundness:** 3
**Presentation:** 3
**Contribution:** 2
**Rating:** 5
**Confidence:** 3

**Summary:**

This paper addresses two key challenges in large-scale image retrieval systems: scalability and efficiency. To enhance scalability, the authors focus on improving the performance of existing foundation models. For efficiency, they propose an effective unsupervised dimension reduction technique.

**Strengths:**

- Overall, the paper is well-motivated and well-written, aiming to solve an important problem in large-scale image retrieval.
- The theoretical aspects are well-developed, though it is unclear if the approximation in Eq. 11 is sufficiently supported. It may benefit from references to statistics or mathematical literature rather than an Arxiv paper in biology.
- Experiments demonstrate the method's effectiveness, showing significant improvement.

**Weaknesses:**

- I am somewhat skeptical about the scenario described in L39-L42; in fact, self- or semi-supervised learning might address this issue (see related works [1, 2]).
- I have some doubts about the novelty of AE-SVC, as it combines several common losses. Additionally, there are four losses in Eq. 5, but no explanation is provided on how they are balanced. Experiments or theoretical support in this regard would be beneficial.
- The experimental results are somewhat limited. Is there additional related work beyond PCA for comparison? Also, I noticed a lack of ablation studies, as only two experiments, shown in Fig.4 and Fig.5, are provided.

[1] eProduct: A Million-Scale Visual Search Benchmark to Address Product Recognition Challenges

[2] V2L: Leveraging Vision and Vision-language Models into Large-scale Product Retrieval

**Questions:**

See weaknesses.

---

### Official Review · Reviewer_mXfW · 2024-11-02

**Soundness:** 2
**Presentation:** 3
**Contribution:** 2
**Rating:** 5
**Confidence:** 3

**Summary:**

This paper discusses two major concerns regarding image retrieval: scalability and efficiency. The concerns are more meaningful when existing retrieval methods use foundation models as feature extractors. To improve the foundation model's performance on retrieval tasks, the authors proposed Autoencoders with Strong Variance Constraints, which enforced three constraints on the learned latent space. Meanwhile, Single-shot Similarity Space Distillation outputs more compact embeddings with flexible dimensions for the model efficiency.

**Strengths:**

The authors focus on major concerns about using the foundation model in image retrieval or even more CV tasks: how to promote foundation model features and balance model performance and efficiency. By aligning teacher and student similarity spaces, the proposed distillation with flexible output dimensions indeed reduces retrieval compute resources.

**Weaknesses:**

1. The motivation of AE-SVC is worth discussing: it encourages the network to learn the normal embedding distribution (mean to 0, std to 1), but gt labels of dataset-specific tasks may biased.
2. The authors claim that they are the first to explore representation learning ideas in image retrieval, considering retrieval is essentially a feature learning task, the soundness is not as remarkable as claimed.
3. AE-SVC is only compared with the standard PCA in Sec.5.2, more competitors (e.g. PCA variants or AnyLoc ) are needed to demonstrate the effectiveness of AE-SVC, if possible.
4. Experiment settings in Fig.4 and Fig.5 are not consistent. Compared with AE-SVC, the results of (SS)2D seems incomplete.

**Questions:**

1. My major concern is whether the learned normal distribution is still effective in the biased target domain, which is common in dataset-specific tasks. Without the prior information of the target domain, it is ambiguous for neural networks to distinguish instances by discriminative features or global features ( discriminate shirt shape or visually more similar image in the example in Fig.2, the former would be right in commerce recommendation system and the latter is more likely to be the output of the common search engine)
2. The settings of Fig.4 and 5 are not consistent, and it is hard to discriminate results in different styles (color, line and dot). It would be better to list the results in tables for better comparison.
3. It is contrary to common sense that the performance of ViT > DINO-Base > DINO-Large in InShop results in Fig.4, but the results of Pittsburgh30K and TokyoVal are normal.
4. Without additional data, student models learned from teachers would get inferior results. Based on AE-SVC (orange) in Fig.5, it seems that applying feature distillation (green) further improves the retrieval performance. It would be better to add an analysis of this point in the experiment.

---

### Official Review · Reviewer_1v5t · 2024-11-06

**Soundness:** 4
**Presentation:** 3
**Contribution:** 3
**Rating:** 6
**Confidence:** 4

**Summary:**

This paper proposes a method for facilitating the image retrieval task. It focuses on solving two challenges, scalability and efficiency, existed in the image retrieval. To tackle scalability, strong variance constraints are introduced to restrict the latent features. The variance of feature in different dimensions could be similar, so that improving the discriminative ability of the latent features. For the efficiency, a Single-shot Similarity Space Distillation method is introduced to generate adaptive size features, and keep the cosine similarity of sample features. The experiments show that the AE-SVC improve the retrieval performance, especially for the foundation models. The (SS)_2D can improve the performance for smaller embedding sizes.

**Strengths:**

1. The paper presents Autoencoders with Strong Variance Constraints (AE-SVC), an unsupervised method. By enforcing orthogonality, mean centering, and unit variance constraints in the latent space, AE-SVC shifts the cosine similarity distribution to be more discriminative. This advancement addresses the scalability issue in current SOTA methods.
2. The authors introduce Single Shot Similarity Space Distillation ((SS)_2D), that preserves the similarity structure of embeddings without requiring retraining for different dimensions. (SS)_2D produces adaptive embeddings where smaller segments maintain high retrieval performance, addressing the impracticality of separate training for varying embedding sizes found in methods like PCA or traditional autoencoders.
3. The paper's two-step approach, first enhancing embeddings with AE-SVC and then applying (SS)_2D, yields significant improvements in retrieval performance across multiple datasets and foundation models. Experimental results show that this method not only accelerates retrieval speed by reducing embedding dimensions but also surpasses the accuracy of existing unsupervised techniques and even the original full-sized embeddings.

**Weaknesses:**

1. The detail of the projection function F in L280 may need more explanations. The size of the embedding \hat{z} is adaptive. The change of the embedding size whether influences the projection F. In other words, there are matched functions for different size m.
2. The figures in Fig. (3,4,5) are small. It is hard to distinguish data points of lower embedding dimensions. It could be better that the quantity results are listed in a table.

**Questions:**

1. During training, does the back propagation happen after traversing the whole dataset or a batch of the images? Would the choice of two ways influence the performance?
2. Does the embedding size reduction influence the cosine similarity variance or other constraints for AE-SVC of the output features?
3. Does the performance be changed if the order of SVC and (SS)_2D switch? Or would the performance be improved if append SVC on the resized embeddings?

---

### Meta-Review · Area_Chair_3QBG · 2024-12-16

**Metareview:**

The paper addresses the scalability and efficiency challenges in image retrieval. Scalability is tackled through strong variance constraints to restrict the latent features, while efficiency is improved via a distillation method that generates adaptive size features. The proposal aims to address problems in pre-trained (or "foundation") models commonly used in retrieval tasks, promoting a balance between performance and efficiency.

Strengths:
- The proposed method shifts the distribution of cosine similarity to be more discriminative.
- The method preserves the similarity structure of embeddings without requiring retraining for different dimensions.
- The paper demonstrates improvements in retrieval performance across multiple datasets and pre-trained models.
- The theoretical aspects are well-developed, with some exceptions.

Weaknesses:
- The contributions in the proposed autoencoder are limited since it relies on common losses. The combination of variance, covariance, and mean-centering constraints is interesting, but their application at the representation level is incremental.
- The learned normal distribution may not be effective in a biased target domain.
- The theoretical analysis is limited to the Gaussian distribution assumption, which does not necessarily hold for embeddings from different visual pre-trained models.
- The experiments require further analysis to better understand the results.
- The baselines used are limited. The paper would benefit from comparisons with other methods.
- The proposed setting is problematic because it shifts the burden from training to testing by requiring learning to be conducted on the test set, which is highly problematic.

Despite the raised concerns, the majority of the reviewers have a positive recommendation given the results presented in the paper.  The minority of the reviewers had concerns regarding the limited technical contribution of the paper.  Nevertheless, I recommend a weak acceptance due to the positive, yet somewhat limited, strengths of the paper.

**Additional Comments On Reviewer Discussion:**

Reviewer 1v5t recognized the two-step proposal as a strength and deemed the improvements significant. The main objections raised were the lack of details in the text and figures, as well as questions about additional details that the authors addressed in their rebuttal. The reviewer mentioned that these concerns were addressed, and maintained a borderline accept rating.

Reviewer mXfW commented that the motivation behind the proposal should be discussed more thoroughly. The reviewer noted that the soundness of the proposal is not as remarkable as claimed, the baselines should be extended since the proposal is only compared against PCA, and the experimental settings are inconsistent. During the rebuttal, the authors responded, and the reviewer was mostly convinced by the response, though some concerns about the limited experimental comparisons and novelty remained. The authors replied again, but the reviewer did not update their assessment.

Reviewer f9Kc found the paper well-motivated with well-developed theoretical aspects and experimental results showing significant improvements. However, the reviewer raised concerns about limited novelty due to the reuse of existing losses and the lack of explanation behind the setup. After the authors' reply and additional experiments, the reviewer maintained that these concerns remained unresolved, citing a lack of clarity and significance in the technical contribution.

Reviewer vMAk remarked on the simplicity and effectiveness of the proposal, as well as the experimental results. The main concern was the lack of theoretical analysis on the Gaussian distribution assumption used in the paper. While additional experimental results strengthened the authors' claims, the reviewer mentioned during the rebuttal that the main concerns regarding the theoretical results were not addressed.

Reviewer Mq3y found the idea well-motivated with a sound methodology, and the experiments supported the claims. The main concern was also the limited novelty due to reliance on existing work. During the rebuttal, most concerns were addressed, but the reviewer mentioned that they would wait until the post-rebuttal discussion to update their score.

Reviewer Mq3y was the only reviewer that replied during the post-rebuttal discussion. The reviewer revised their score and provided a detailed description of the reasons for their borderline decision, requesting that the authors update the paper to include most of the discussed issues.

While Reviewer f9Kc had several exchanges with the authors, the main issue remaining was the technical novelty. While I agree that the proposal is incremental, the additional results showing that the proposal works is a positive signal. The other negative reviewer, f9Kc, had concerns about the lack of experimental results and thorough evaluations. Overall, the paper appears to be a borderline case, with the majority of reviewers giving a positive but weak acceptance signal based on the results, while the other two reviewers give a weak rejection signal based on the lack of technical novelty. Given this imbalance, I suggest accepting the paper following the majority of reviewers' recommendations.

---

### Decision · Program_Chairs · 2025-01-22

Accept (Poster)